# The Adhesome Network: Key Components Shaping the Tumour Stroma

**DOI:** 10.3390/cancers13030525

**Published:** 2021-01-30

**Authors:** Pinelopi A. Nikolopoulou, Maria A. Koufaki, Vassiliki Kostourou

**Affiliations:** Biomedical Sciences Research Centre “Alexander Fleming”, Institute of Bioinnovation, 34 Fleming Str., 16672 Vari-Athens, Greece; nikolopoulou@fleming.gr (P.A.N.); koufaki@fleming.gr (M.A.K.)

**Keywords:** tumour microenvironment, adhesome, focal adhesion sites, endothelial cells, mural cells, cancer-associated fibroblasts, immune cells, tumour stroma

## Abstract

**Simple Summary:**

Tumours are not formed only by malignant cells but contain many other cell types, including endothelial and mural cells of blood vessels, immune cells and cancer-associated fibroblasts. These host cells, together with extracellular matrix, form the tumour stroma. Tumour growth and metastasis depends on interactions between cancer cells and tumour stroma. Cell adhesion to extracellular matrix is essential for tissue growth and homeostasis and is deregulated in many pathological conditions, including cancer. This review highlights the vital role of cell adhesion in malignancy and describes how adhesion components regulate tumour stroma responses and control cancer development.

**Abstract:**

Beyond the conventional perception of solid tumours as mere masses of cancer cells, advanced cancer research focuses on the complex contributions of tumour-associated host cells that are known as “tumour microenvironment” (TME). It has been long appreciated that the tumour stroma, composed mainly of blood vessels, cancer-associated fibroblasts and immune cells, together with the extracellular matrix (ECM), define the tumour architecture and influence cancer cell properties. Besides soluble cues, that mediate the crosstalk between tumour and stroma cells, cell adhesion to ECM arises as a crucial determinant in cancer progression. In this review, we discuss how adhesome, the intracellular protein network formed at cell adhesions, regulate the TME and control malignancy. The role of adhesome extends beyond the physical attachment of cells to ECM and the regulation of cytoskeletal remodelling and acts as a signalling and mechanosensing hub, orchestrating cellular responses that shape the tumour milieu.

## 1. Introduction

Malignant cells are the main driving force of tumour formation and growth, yet they do not manifest the disease single-handed. It is now increasingly accepted that noncancerous cells located in the tumour niche are involved in the hallmark capabilities of cancer [1]. Key cellular players in the tumour stroma are vascular cells comprising blood and lymphatic vessels (endothelial cells and mural cells), cancer-associated fibroblasts (CAFs), and infiltrating immune cells. Additional stromal cell types of solid tumours include mesenchymal stem cells, adipocytes and neurons [2,3]. The complex contributions of these allegedly normal, constituent cells together with noncellular components, namely the extracellular matrix (ECM) and soluble factors, are collectively known as the tumour microenvironment (TME) (Figure 1).

As in normal tissue, all cells in the TME are attached and interact with ECM at specific sites, called cell–matrix adhesions. These structures are composed by the integrin family of ECM receptors and an intrigue protein network called adhesome that is directly linked to the cell cytoskeleton [4,5,6]. Adhesion sites physically attach the cell to its microenvironment and maintain cell homeostasis. Depending on the protein composition and arrangements within the adhesome, cell–matrix adhesions have distinct architecture, localisation and lifespan [7]. These dynamic characteristics enable adhesion sites to rapidly change cell morphology and modulate cell spreading to facilitate cell migration and invasion. Furthermore, cell–matrix adhesions co-ordinate cellular responses to extracellular stimuli through crosstalk with intracellular signal transduction pathways, hence influencing cell survival, proliferation and differentiation. Emerging data highlight the role of mechanical cues in regulating cell function. In this aspect, adhesome proteins constitute critical players in both sensing and transducing mechanical forces to cell cytoskeleton. Adhesome members include more than 232 adhesion-related proteins broadly divided into structural/adaptor proteins (such as talin, vinculin, ILK-PINCH-parvin, paxillin, zyxin, tensin, α-actinin, filamin and KANK) and signalling proteins (for example FAK, Pyk2, Src, RhoGTPases and myosin light chain kinase). These proteins could constitute intrinsic components of cell–matrix adhesions or transiently associate with the adhesion network, facilitating the coordination of cell behaviour [8,9,10,11,12]. Finally, adhesome components connect with different actin structures (stress fibres, cortical actin, lamellipodia, filopodia) and microtubules, to stabilise cell shape and facilitate cell migration and invasion.

Several experimental approaches, ranging from genomics, proteomics and bioinformatics, to classical biochemistry, cell biology and model organisms (fly, worm, mouse), have started to decipher both the composition and the role of adhesome members in orchestrating cell function in physiology and pathology. In this review, we elaborate on the functional impact of adhesome proteins in cancer, focusing on host cells located into the tumour niche: endothelial cells (ECs), mural cells, CAFs and immune cells (summarised in Table 1).

To delineate the involvement of critical adhesome components on tumour stroma, we refrain from reviewing the role of adhesion molecules in cancer cell properties. Instead, we present the current understanding of how adhesome components control tumour stroma attained from cell-specific gene deletion and inhibition studies in different tumour settings. When such knowledge is missing, we discuss findings from genetic animal studies that provide valuable insight into stroma cell-specific roles of adhesome members under physiological conditions (summarised in Table 2). Since there are excellent reviews on the role of integrins on TME [52,53,54,55,56], we focus on the function of intracellular adhesome components in the tumour stroma. For clarity, we classify the adhesome proteins into three distinct categories, namely the adhesome signalling, the adhesion organisation and the actin regulatory layer (Figure 2). This classification has limitations because it does not take into account the dynamic crosstalk and the combined multiple roles of many adhesome members. We believe, though, that this simplifying approach will provide a clearer framework to emphasise the essential role of cell–matrix adhesions on the TME.

## 2. Adhesome Function in Vasculature

Blood vessels are composed of ECs, forming the inner surface of vessels, and perivascular mural cells (pericytes and smooth muscle cells) surrounding the vascular network. Under physiological conditions, blood vessels supply tissues with nutrients and oxygen and discharge metabolic waste products [123]. Cancer cells co-opt blood vessels and favour angiogenesis—the development of new blood vessels from pre-existing ones—to promote tumour growth and obtain routes for metastasis [124,125]. Owing to deregulated angiogenic processes, the tumour-associated vasculature has abnormal structure and function [126,127]. Unlike normal vessels, tumour vessels lack hierarchical organisation and exhibit irregular branching, aberrant density, shunting, incoherent blood flow and leakiness [128], establishing a chaotic vascular network with impaired functionality.

### 2.1. Endothelial Cell Adhesome

ECs are strategically positioned to influence the TME because of their ability to proliferate and migrate to form new blood vessels, fuelling tumour growth. Moreover, ECs serve as gatekeepers, enabling circulating factors and immune cells to penetrate the tumour environment and tumour cells to enter the circulation in order to form metastasis. They contribute to cancer development through communication with other stroma components and cancer cells. ECs are also critical drivers of tumour ECM remodelling. Endothelial adhesome has emerged as a crucial player of tumour progression because it controls both cell–cell junctions and cell–ECM attachment, thus influencing cellular crosstalk, angiogenesis and leakage.

#### 2.1.1. Adhesome Signalling

The most studied adhesome signalling member both in physiology and cancer is focal adhesion kinase (FAK). FAK is abundant in developing blood vessels suggesting that FAK may play a role in angiogenesis [129]. Indeed, overexpression of FAK in ECs enhanced angiogenesis during wound healing and recovery from hind limb muscle ischemia [130] and endothelial-specific deletion of FAK led to embryonic lethality between embryonic day (E) 10.5 and 11.5 due to defective angiogenesis [57,58]. The critical role of FAK in tumour growth and angiogenesis was established in later studies by inducible deletion of FAK in tumour endothelium [13,14]. Tarova et al. showed that the endothelial-specific ablation of FAK in adult mice reduced tumour growth by inhibiting tumour angiogenesis. Mechanistically, endothelial FAK deficiency reduced cell migration and proliferation and enhanced apoptosis, resulting in a less dense vascular network both in tumour and in the developing retina [13]. Similarly, Lee et al. showed that deleted FAK from adult ECs reduces glioma growth and decreases tumour vascular dilation, tortuosity, and permeability by stabilising brain EC junctions and astrocyte feet interactions [14]. FAK was shown to be dispensable for VEGF-induced angiogenesis in adult mice due to compensatory Pyk2 expression [60]. However, endothelial FAK ablation did not change Pyk2 levels in embryonic or cancer vascular development [13,14,59,131,132]. In addition to the importance of endothelial FAK expression in angiogenesis and tumour growth, endothelial-specific deletion of FAK enhanced tumour sensitivity to DNA-damaging drugs by increasing inflammatory cytokine production, indicating that endothelial FAK could be targeted in combinatorial anticancer treatment [133].

Several studies have shown that pharmacological inhibition of FAK results in decreased tumour growth [134,135,136,137]. A dose-dependent effect of FAK inhibition in tumour growth and angiogenesis, however, was demonstrated in a study using FAK-heterozygous mice that displayed increased tumour growth [15]. These findings raised the necessity for more careful characterisation of FAK function before clinical application. To delineate FAK function in vivo, different mutant mice were generated targeting the FAK kinase domain, the autophosphorylation tyrosine site Y397 (blocking Y397F or mimic Y397E phosphorylation), and the Src-phosphorylation site Y861 (blocking Y861F) [16,17,18,59,131,132,133,138]. Embryos that lack FAK kinase activity or FAK-Y397 phosphorylation died at E13.5–15.5, exhibiting haemorrhages, oedema and vascular remodelling defects [59,131,132]. In agreement with embryonic studies, FAK kinase activity in ECs was necessary to induce phosphorylation in tyrosine residue Y658 of VE-cadherin and thus destabilisation of adherens junctions (AJ) [16,17,59]. Inhibiting FAK kinase activity in tumour endothelium decreased vascular leakage, tumour growth and metastasis [16,17]. Interestingly, an endothelial-specific FAK Y397E mutant, that mimics the phosphorylation of FAK Y397, was sufficient to restore VE-cadherin Y658 phosphorylation in tumour ECs and induce tumour vascular permeability in FAK kinase-deficient mice [17]. Consistently, blocking FAK phosphorylation at tyrosine Y397 in ECs inhibited tumour growth and angiogenesis [18]. The endothelial-specific phosphorylation of FAK at tyrosine residue-861 impairs the VEGF-dependent angiogenesis at early stages of tumour development, but this effect is not sustained in late-stage tumours [18]. Taken together, these data demonstrate that an intricate balance of FAK regulation is required to obtain the desired anticancer outcome.

#### 2.1.2. Adhesion Organisation

Despite the indispensable role of many adhesome members in cell adhesion organisation and their effect in cancer cell motility and survival, genetic evidence for their function in tumour-associated ECs is mainly unexplored. Indirect evidence exists for the involvement of kindlin2 and integrin linked kinase (ILK) in tumour angiogenesis [36,139]. Prostate tumours growing in kindlin2 heterozygous mice were smaller and developed significantly less tumour blood vessels compared to control littermates [36]. Consistently, the heterozygous expression of kindlin2 reduced VEGF responses in a Matrigel angiogenesis assay in vivo [36]. Moreover, kindlin2 partial reduction caused defective basement membrane and pericyte coverage, resulting in impaired vascular maturation and increased vessel leakage [36]. Later studies further showed that kindlin2 regulates endothelial barrier function [140]. These data raise the possibility that kindlin2 could affect the formation of metastasis.

ILK is an essential mediator of integrin signalling, regulating cytoskeletal organisation and dynamics [141] and cancer cell survival. In thyroid tumour xenografts, pharmacological inhibition of ILK triggered apoptosis in both tumour cells and ECs, thus inhibiting tumour volume and angiogenesis [40]. The same pharmacological approach was used to suppress ILK expression in glioblastoma cells resulting in smaller tumours and reduced vessel density [42,142]. In an orthotopic model of human prostate cancer, siRNA-mediated depletion of ILK reduced tumour growth and angiogenesis by inhibiting HIF1a and VEGF expression [139]. In a paracrine action, ILK knockdown in melanoma cells decreased angiogenesis by lowering the proinflammatory cytokine IL-6 [143]. Pharmacological inhibition of ILK also resulted in decreased VEGF-induced EC invasion into Matrigel plugs in vivo [139]. Collectively, these data indicate that ILK could affect tumour growth by regulating angiogenesis. In pancreatic cancer xenografts, however, pharmacological ILK inhibition reduced tumour growth without affecting tumour angiogenesis [41]. Direct evidence for the significance of ILK expression in tumour-associated ECs is still missing. Accumulating in vitro data indicates a significant function of ILK in ECs. ILK regulates EC migration and responses to angiogenic factors VEGF and EGF and is necessary for integrin α_5_β_1_-mediated adhesion and capillary tube formation in vitro [139,144,145]. Furthermore, ECs derived from embryonic stem cells lacking ILK expression could not form vessel-like structures due to perturbed organisation of microtubules and cortical actin filaments and defective caveolin-1 localisation [144]. In vivo studies further support the vital role of ILK in endothelial physiology. Endothelial-specific deletion of ILK in mice results in embryonic lethality due to defective placenta vascularisation [87]. Inducible EC deletion in postnatal mice decreased sprouting of the retinal vasculature, reduced EC proliferation and disrupted the blood–retina barrier causing increased vascular leakage [88]. Therefore, deciphering whether and how the endothelial expression of ILK affects tumour growth and angiogenesis could support ILK as a target for anticancer therapies.

Besides ILK, talin is a key adhesome organiser because it controls the cell–ECM attachment by activating integrins and directly binding to the actin cytoskeleton. In addition, talin’s unique mechano-sensitive structure serves as a scaffold for the recruitment of numerous adhesome molecules. Endothelial-deletion of talin1 results in embryonic lethality due to vascular defects, while inducible ablation of talin1 in adult endothelium causes severe haemorrhaging in the intestinal vasculature followed by death [77,78]. Recent studies also showed that integrin activation by talin1 plays an important role in tumour angiogenesis. Endothelial-specific expression of an integrin activation talin mutant (talin1 L325R) decreased subcutaneous melanoma tumour growth and angiogenesis [39]. In line with the role of talin in endothelial junctions, vinculin, a close interactor of talin, is required for the remodelling of AJs [146]. As a response to different barrier-disruptive or barrier-enhancing agonists, vinculin localisation at AJ affects the actomyosin dynamics and thus the intracellular forces that are essential for the integrity of EC barrier [147]. These findings make vinculin and talin key candidates for regulating tumour vascular permeability and metastasis, a notion that remains to be investigated.

#### 2.1.3. Actin Regulatory Layer

Cytoskeletal dynamics are crucial for cell shape maintenance and cell membrane extensions during cell migration and invasion. Many adhesome components are involved in linking the adhesion protein network to actin and tubulin cytoskeleton. How these proteins influence tumour stroma is mostly unexplored. Rho-GTPases family are the widely accepted master controllers of actin dynamics. The best-characterised members are Cdc42, which affects filopodia formation, Rac which guides lamellipodia development, and RhoA, which promotes stress fibres [148,149]. Although extensively studied in cancer cell migration, invasion and metastasis, the stroma cell-specific roles of Rho-GTPases family are less understood [150]. D’Amico and colleagues demonstrated that deleting Rac1 specifically from tumour ECs did not alter tumour growth and angiogenesis in wild-type mice but significantly decreased tumour vascular formation and inhibited tumour development in mice lacking integrin β_3_ expression [48], indicating a context-dependent role of Rac1 in tumour ECs. Another study showed that silencing Rac1 expression in the whole tumour xenograft, reduced tumour growth and angiogenesis and inhibited VEGF-induced neovascularisation of Matrigel plugs in vivo [49]. During development, endothelial deletion of Rac1 resulted in embryonic lethality caused by defects in heart development, early vasculogenesis and lymphatic-blood vessel separation [43,151]. In developing retina vasculature, endothelial deletion of Rac1 decreased vessel branching and EC invasion necessary to form the deeper plexus [108]. Taken together, these findings indicate an intricate involvement of Rac1 in tumour development, driven by EC-specific expression.

Another member of the Cdc42 subfamily, RhoJ, is essential for pathological angiogenesis [152,153]. RhoJ has distinct vascular expression pattern which is regulated by the endothelial transcription factor ERG and is required for EC migration, proliferation, and tube formation [154,155]. Although mice with constitutive deletion of RhoJ are viable, comprehensive examination revealed reduced vessel branching during embryogenesis and a delay in retinal vascular development due to impaired EC migration [120]. The effect of RhoJ deficiency was stronger in tumour settings, where it drastically decreased tumour angiogenesis and growth [50]. These findings were confirmed by endothelial-specific targeting of RhoJ, which suppressed blood vessel formation and disrupted tumour vascular integrity and function [51]. These differences in developmental and pathological angiogenesis could be caused by compensatory mechanisms acting in different contexts. Similar compensation could exist between various members of the RhoA family since endothelial deletion of RhoA did not affect developmental angiogenesis [117].

Besides the Rho-GTPases family, other actin-binding proteins could have an EC-related impact in cancer progression. Endothelial-specific deletion of filamin A decreased tumour angiogenesis and fibrosarcoma development [47]. Similarly, depletion of filamin B in HUVECs prevented EC migration and VEGF-induced tube formation in vitro [156] while constitutive deletion of filamin B in vivo caused nonlethal defects in microvascular patterning [103]. Collectively, these findings imply an important role of filamins in regulating tumour growth and angiogenesis.

#### 2.1.4. Emerging Adhesome Players in Tumour Endothelium

Evidence from genetic studies have demonstrated a vital role of other adhesome components in endothelial biology, making them intriguing targets to be further explored in tumour settings. For example, parvin family proteins, α- parvin and β-parvin, form a tri-partite complex called IPP, with ILK and particularly interesting new cysteine-histidine-rich protein (PINCH) proteins. Endothelial-specific deletion of α-parvin reduced angiogenic sprouting and impaired vessel stability during late embryonic developmental stages [91]. Similar defects were observed in postnatal retina vascular formation [91]. Moreover, mice with endothelial deletion of both α- and β-parvins die earlier than single α-parvin deficient mice, indicating a partial compensatory function between parvin proteins [92]. Detailed examination of the embryonic vasculature revealed an abnormal vascular network with reduced branching, dilated vessels, balloon-type endothelial clusters and discontinuous basement membrane leading to extensive haemorrhaging. Parvin deficient vessels had defects in endothelial cell–cell junctions, apical-basal polarity and pericyte–vessel coverage [92]. All these defects could manifest in tumour vasculature but further studies are required to establish the role of parvins in tumour ECs.

Moreover, the putative role of tensins, a family of intracellular adhesion proteins that links integrins to actin filaments in cancer, stems from in vivo studies. Specifically, tensin1 deficiency in mice, although viable, has been shown to reduce angiogenesis both in vivo Matrigel assay and in ex vivo aortic ring assay [96,157]. This effect is mediated by decreased EC proliferation, migration and tube formation caused by downregulation of RhoA activity [96]. Thus, one can postulate that tensins could affect tumour angiogenesis. Another adhesome member that is closely linked to actin is zyxin [158]. Zyxin could affect tumour haemorrhaging and growth because it has been shown to regulate the exocytosis of endothelial Weibel–Palade body (WPB) and the secretion of von Willebrand factor (vWF) [159]. Interestingly, zyxin is upregulated upon shear stress and translocates from cell–matrix adhesions to the nucleus where it affects gene expression [160], indicating alternative ways to influence tumour development. Besides zyxin, another structural component of adhesion sites that can translocate to the nucleus and affect transcription is the family of four and a half LIM domain protein (FHL). In vitro studies propose that FHL2 has an antiangiogenic role because it can bind and inhibit HIF1a transactivation and VEGF transcription [161,162] and impairs EC migration and survival [163,164]. However, angiogenesis in the ischemia hind limp model and during recovery from corneal injury is not elevated but decreased in FHL2 knockout mice [165,166]. More studies are needed to decipher the involvement of FHL family proteins in tumour vascular development and cancer progression.

Another adhesion component that can influence tumour growth by affecting both vessel formation and permeability, is paxillin. Paxillin interacts with many kinases, including FAK, Src, Erk, Akt, PKA, and tyrosine phosphatases [167,168] and depending on its phosphorylation status can induce the assembly or the disassembly of adhesion sites [169]. Moreover, paxillin can regulate cell migration and invasion by controlling the spatiotemporal dynamics of Rho-GTPase family proteins Rho, Rac and Cdc42 through the recruitment of several GEFs and GAPs and interactions with the GIT/PIX/PAK complex [170,171]. Consistent with this, paxillin knockdown using siRNA enhanced EC migration and invasion in vitro and increased retina vascular sprouting in vivo by reducing neuropilin2 expression [32]. Interestingly, tumour soluble factors decreased endothelial expression of both paxillin and neuropilin2 and increased angiogenesis in a Matrigel assay in vivo [32], indicating an antiangiogenic role for paxillin in tumour setting. Additional to modulating angiogenesis, paxillin could affect cancer development by its established role in regulating endothelial barrier function [172,173,174]. The phosphorylation of paxillin in tyrosine-31 and -118 sites leads to destabilisation of VE-cadherin and enhanced pulmonary vascular permeability in LPS-mediated lung injury [173]. Nevertheless, in the presence of barrier enhancing factors such as sphingosine-1-phosphate (S1P) and hepatocyte growth factor (HGF), paxillin reinforces endothelial barrier through the formation of lamellipodia [174]. Therefore, it would be informative to investigate the effect of endothelial deletion of paxillin in tumour vascular permeability and metastasis.

Indications for a role of Cdc42 subfamily in tumour stroma stems from developmental studies showing that endothelial deletion of Cdc42 results in embryonic lethality with impaired vascular function [104,105]. Specifically, endothelial loss of *cdc42* reduced EC migration and survival by decreasing the surface availability of VEGFR2, and thus angiogenic responses [105]. A more detailed examination revealed that endothelial deletion of *cdc42* disrupted lumen formation during embryonic angiogenesis and reduced vascular sprouting and filopodia formation in the developing retina. These defects were caused by a disorganised actin cytoskeleton, loss of cell adhesion and endothelial polarity [104]. Besides the Rho family GTPases, other actin regulators could influence tumour vascular function. Specifically, VASP is upregulated during capillary morphogenesis in vitro [175] and it regulates the tethering of actin filaments during the formation of endothelial cell–matrix and cell–cell contacts [176]. Several studies, also, have demonstrated an essential role of VASP in maintaining endothelial barrier function in response to several stimuli that also affect tumour vasculature, including nitric oxide (NO) and hypoxia [177,178,179]. Due to functional compensation between the Ena, VASP and Ena/VASP-like proteins, single knockout mice have no major defect. The triple knockout mouse, though, is lethal and displays impaired endothelial cell junctions, leading to oedema and vascular leakage [97], indicating an important and yet undiscovered function in tumour vascular permeability and metastasis.

### 2.2. Mural Adhesome

Mural cells, comprising vascular smooth muscle cells (vSMCs) and mesenchymal-like cells called pericytes, are critical constituents of the TME [180]. In the TME, both vSMCs and pericytes are loosely associated, enabling vessel leakage [127,181]. Emerging data suggest a critical role of pericyte adhesion in regulating tumour growth and metastasis. In particular, deletion of integrin α_v_β_3_ specifically from mural cells increased tumour development and metastasis by paracrine survival and tumour promoting signals without causing major defects in the tumour vasculature [182]. Another study showed that metastatic cells exploit integrins, ILK and YAP to dislodge pericytes, spread on capillaries, and form colonies in distant organs [183]. Despite these findings, very little is known about how adhesome components intrinsically influence mural function in cancer.

#### 2.2.1. Adhesome Signalling

Recent studies using genetic deletion approaches have addressed the role of mural FAK in tumour angiogenesis and cancer development. In particular, mural FAK deficiency enhanced tumour growth and angiogenesis in syngeneic subcutaneous mouse models and spontaneously arising RIP-Tag2 pancreatic tumours [19]. In both models, FAK deficiency weakened the association of pericytes with tumour blood vessels and increased tumour angiogenesis. Mechanistically, the loss of pericyte FAK enhanced the expression of the proangiogenic and tumorigenic cytokine Cyr61, via a Gas6/Axl axis, driving tumour progression. These results were corroborated by human melanoma studies, where a high correlation was shown between tumour size and loss of pericyte-FAK [19]. As in ECs, pericyte FAK phosphorylation appears to have distinct roles in tumour growth and angiogenesis. Whereas phosphorylation of mural FAK at Y397 does not seem to affect angiogenesis and tumour growth, the phosphorylation at Y861 is important for blocking vessel regression and enhancing tumour survival [20].

#### 2.2.2. Adhesion Organisation

The impact of adhesome structural components in mural function during cancer development has not been studied. Nevertheless, evidence from mural specific genetic ablation of key adhesome members advocates a critical role in tumour vascular function. A direct paradigm constitutes the mural ablation of ILK, which results in the defective formation of the vessel wall and embryonic lethality. Mechanistically, ILK deficiency enhances phosphorylation of myosin light chain (MLC) through the activation of Rho/ROCK signalling pathway and enhances vSMC contraction both in vitro and in vivo [89]. The same Rho/ROCK signalling pathway is also activated in the absence of α-parvin. Deletion of α-parvin in mice is embryonic lethal with defects in vascular remodelling and reduced vSMC/pericyte spreading on blood vessels due to hypercontractility [93]. Elevated contractility was also observed in pericytes overexpressing talin in vitro. This contractile phenotype was shown to be dependent on calpain-mediated signalling, as calpain pharmacological inhibition and expression of a talin mutant that is resistant in calpain cleavage reversed this effect [184]. In contrast, zyxin-null vSMCs display reduced responses to contractile agonists and exhibit enhanced proliferation, migration and invasion and are resistant to stress-induced apoptosis [185]. More investigation is needed to address the functional role of these key adhesome members in tumour pericyte function and cancer development.

## 3. Adhesome Function in CAFs

Fibroblasts are spindle-shaped cells of mesenchymal origin and have a pivotal role in tissue repair through a reversible transition to activated fibroblasts (also termed as myofibroblasts) [186]. CAFs represent one of the most dominant components of the tumour stroma in several malignancies including breast [187], liver [188], gastrointestinal [189], and pancreatic [190] cancer. In contrast to normal fibroblasts, the majority of CAFs remain in an “activated” state characterised by enhanced contractility, proliferation and ECM deposition [191,192]. Advances in single-cell scale approaches revealed that CAFs are a heterogeneous population with functional diversity deriving from multiple origins [193,194]. Currently, unique markers for CAFs have not been defined, and therefore CAFs are mainly recognised by the expression of myofibroblast proteins, such as a-smooth muscle actin (aSMA), fibroblast activation protein (FAP), fibroblast-specific protein 1 (FSP1), and vimentin [195]. Several studies have revealed that CAFs have either a pro- or an antitumorigenic potential within the TME, implying a dual role of CAFs in cancer progression [196].

### 3.1. Adhesome Signalling

Several studies show that CAFs display altered adhesion-mediated signalling that favours their activation and their protumorigenic capacity. For example, microarray gene- profiling in cultured CAFs and matched normal fibroblasts from nonsmall cell lung carcinoma (NSCLC) patients, together with data of differentially expressed stromal/CAF genes from various solid tumours revealed significant enrichment in focal adhesion pathways [197], with integrins and FAK posing central nodes [197,198]. Moreover, primary CAFs from NSCLC patients displayed elevated expression of integrin β_1_ and FAK-Y397 phosphorylation, driving proliferative responses to matrix rigidity and differential accumulation in squamous cell carcinoma vs. adenocarcinoma, in vivo [199]. Similarly, patient-derived oral squamous cell carcinoma (OSCC) CAFs overexpress FAK and siRNA mediated FAK depletion in CAFs reduced monocyte chemoattractant protein 1 (MCP1) production and decreased cancer cell invasion in vitro [200]. Recently, it was reported that FAK kinase activity regulates the TGFβR2 recycling and hepatic stellate cell (HSC) activation to tumour-promoting CAFs in vitro and in mice [23]. In other studies, chemical inhibition of FAK resulted in tumour stasis by reducing stroma proliferation and limiting the presence of FAP^+^ and aSMA^+^ CAFs [21,22]. Moreover, FAK activity in CAFs was recently found to drive tumour metastasis and augment an immunosuppressive TME [25,26]. Taken together, these studies provide evidence for a role of FAK in mediating the tumour-promoting actions of CAFs in vivo. Strikingly, a recent study has further addressed the direct involvement of FAK in CAF properties and discovered a tumour suppressive role for CAF-FAK by regulating cancer cell metabolism [24]. Specifically, conditional deletion of FAK from FSP1-expressing CAFs increased tumour growth in both orthotopic and spontaneous breast cancer mouse models. The enhanced tumour development in FSP-Cre^+^;FAK^fl/fl^ knockout mice was caused by altered CAF chemokine production and paracrine signalling, which increased malignant cell glycolysis. These findings were corroborated by clinical observations that low FAK stromal expression is positively correlated with poor overall survival in human breast and pancreatic patients [24].

### 3.2. Adhesion Organisation

Besides FAK, recent studies revealed that kindlin2 is necessary for bladder CAF activation and pancreatic stellate cell (PSC) proliferation, migration and cytokine production [37,201]. Kindlin2 siRNA-mediated ablation in isolated human PSCs decreased pancreatic cancer cell proliferation and migration in vitro and growth of tumour/CAFs co-implantation in athymic mice [37]. Similarly, kindlin2 silencing suppressed CAF-mediated cancer cell survival, migration and epithelial to mesenchymal transition [201]. Altered adhesion properties are crucial not only for the differentiation of fibroblasts into CAFs and their migratory capacity but also for the production and organisation of ECM, that governs tumour rigidity and response to treatment [202]. Besides the role of integrins, emerging data indicate an involvement of other adhesome components in CAF-mediated ECM remodelling. For example, CAFs exhibit an increased number of adhesion sites and higher turnover rates of large vinculin-containing adhesions that enhance contractility and traction force generation necessary for aligning fibronectin (FN) fibres and creating migratory tracks for cancer cells [203]. A separate study implicated talin1 and kindlin2 for the activation of β_1_ integrins and the cooperation with the DDR2 collagen receptor to enable CAF-mediated collagen matrix assembly and linear organisation [204].

### 3.3. Actin Regulatory Layer

Given the essential role of actomyosin cytoskeleton in eliciting CAF contraction, actin-related proteins have emerged as critical effectors of CAF function. Microarray analysis in CAFs isolated from patients with distinct breast cancer molecular types revealed that CAFs from Her2-positive tumours had significant upregulation of integrin and actin-related pathways including Rho GTPases and kinases [205]. A separate study showed that blocking the mechanical activity of CAFs by inhibiting ROCK kinase decreases vascularisation in a 3D assay, indicating an additional way by which adhesome components regulate CAF-tumour promoting function [206]. Furthermore, a key role for VASP in HSC activation into myofibroblasts and CAF-related tumour progression has been shown in vitro and in vivo experiments. Specifically, siRNA knockdown of VASP in mouse embryonic fibroblasts failed to induce tumour progression and an αSMA-enriched microenvironment in a subcutaneous mouse tumour/CAF coengraftment model [45]. Another study, showed that VASP expression is increased in HSCs associated with liver metastasis both in experimental mice and cancer patient samples and this is required for TGFβ signalling, differentiation of normal HSCs to myofibroblasts and CAF paracrine function [46].

In addition to VASP, several studies have established that increased expression of a long palladin isoform is mostly restricted in the tumour stroma, and elevated palladin levels in CAFs are associated with poor prognosis and unfavourable chemotherapeutic efficacy of pancreatic and renal cancer [207,208,209,210]. It has been shown that palladin has a prominent role in mechanosensing, and adhesion dynamics of pancreatic CAFs [211,212]. Furthermore, palladin can control the activity of Cdc42 and increase the formation of adhesive membrane protrusions, called invadopodia, that can remodel ECM and promote tumour development in vivo [44,213]. Taken together, these data suggest that palladin is a key regulator of CAF tumour-supportive properties.

## 4. Adhesome Function in Immune Cells

Tumour-infiltrating immune cells (TICs) contribute to many if not most hallmarks of cancer [1]. The “proinflammatory” and cytotoxic activities of immune cells lead to cancer cell destruction, while distinct leukocyte subsets acquire inhibitory functions that benefit tumour growth and immune evasion. The abundance and constitution of TICs vary considerably depending on the tumour type [3]. The spatial organisation of immune infiltrates within the TME is now increasingly recognised as a dynamic nexus of chemokine-driven cell motility rather than simple segregation into peritumoral, stromal and intratumoral location [214]. Here, we classify TICs into two main categories according to their recruitment in innate (macrophages, natural killer cells, monocytes, myeloid-derived suppressor cells, neutrophils, dendritic cells or DCs), or adaptive (T and B lymphocytes) immune responses to review the role of adhesome components in immune TME composition and function.

### 4.1. Innate Immune Cell Adhesome

Inside the TME, tumour-associated macrophages (TAMs) are the most prevalent immune constituent. They serve as a mix of tissue-resident and monocyte-derived cells that are involved in every step of malignant progression, from the primary tumour formation to metastatic colonisation [215,216]. In a general term, two fundamental phenotypes of TAMs have been documented, termed as the M1 (“classically” activated, proinflammatory, tumouricidal) and the M2 (“alternatively” activated, anti-inflammatory, tumour-promotive) type, but accumulating evidence has also described plasticity between TAM states and multiple distinct phenotypes combining features of both M1 and M2 polarisation [217]. Besides TAM, other leukocyte populations including recruited monocytes, tumour-associated neutrophils (TANs), myeloid-derived suppressor cells (MDSCs), dendritic cells and natural killer (NK) cells are critical components of innate tumour responses. Current research has now begun to scratch the surface of adhesome-mediated regulation within the immune TME.

#### 4.1.1. Adhesome Signalling

Early in vitro studies showed that FAK and the related Pyk2 signalling are vital for macrophage responses [218,219], including differentiation [220], cytokine production [221,222], migration [66,223], and phagocytosis [224,225]. Besides FAK and Pyk2, paxillin phosphorylation has been implicated in cytokine-induced neutrophil activation and macrophage phagocytosis, in vitro [226,227,228]. However, the impact of paxillin signalling on individual innate immune subsets in vivo and the regulation of TME within different tumour types remains to be explored.

Genetic ablation of FAK and Pyk2 in myeloid lineages demonstrated their essential role in macrophage recruitment during inflammation [61,229]. Likewise, genetic studies in neutrophils revealed that FAK and Pyk2 regulate neutrophil chemotaxis, degranulation and host-killing responses [62,67]. Pyk2 kinase activity has also been involved in regulating chemokine-driven DC motility in vitro [230,231] and a recent study attempted to delineate how Pyk2 activity defines the ratio of monocyte subsets at steady state [68]. Taken together, these studies suggest that FAK/Pyk2 signalling could play an important role in tumour innate responses.

Indeed, inhibiting FAK kinase activity in murine models of breast and pancreatic cancer reduced tumour growth and limited CD45^+^ immune cells [27] and TAM infiltration [21,22,28], indicating a tumour-promoting FAK function. Accordingly, high levels of FAK phosphorylation within tumour tissue have been correlated with elevated granulocyte recruitment in human pancreatic cancer and FAK inhibition decreased tumour penetration of both monocytic MDSCs and polymorphonuclear/granulocytic MDSCs [22]. However, these studies were mainly focused on cancer cell-mediated effects of FAK inhibition, and a more detailed examination is needed to dissect the effects of FAK inhibition on individual innate immune subsets within different tumour types. A recent study showed that mechanical stretch preconditioning of macrophages activates FAK and induces M1 polarisation. Intratumoural injection of mechanical stretched macrophages decreased tumour growth in vivo [232], suggesting an antitumour role for FAK in macrophages. In agreement with a multifaceted action of FAK in the tumour stroma, myeloid cell-specific deletion of FAK in *LysM-Cre* mice with spontaneously developed mammary carcinomas was associated with both pro- and antitumorigenic functions depending on the stage of malignancy [29]. In particular, FAK depletion in myeloid cells retards early tumour progression, but it accelerates outgrowth after primary tumour formation by reducing tumour-infiltrating NK cell abundance [29].

NK cells are innate lymphocytes with antitumour cytolytic activity [233]. Previous in vitro studies showed that Pyk2 kinase activity along with paxillin activation contributes to NK cytotoxic efficiency [234,235,236,237] and inhibition of Pyk2 activity decreased NK trans-endothelial migration [69], indicating that Pyk2 and paxillin signalling could potentially mediate NK cell extravasation and tumour infiltration limiting tumour progression. Although the increased density of intratumoral NK cells has been associated with a beneficial prognosis [238,239,240], immune cell recruitment in the tumour niche does not necessarily coincide with functionality. Tumours discover ways to evade the cytotoxic function of NK cells and impel them to enter an anergic state [241]. Therefore, further in vivo studies are required to determine the functional significance of adhesome signalling in NK-mediated tumour responses.

#### 4.1.2. Adhesion Organisation

Additionally to the role of adhesion scaffolding proteins, including talin, vinculin and α-actinin on phagosome structure [226,242], and function [243,244], adhesion organisation is critical for podosome assembly and innate immune cell migration [245,246,247]. Apart from the facilitation of cell migration, podosomes in DCs have been proposed to participate in antigen representation [248]. In agreement with this concept, genetic deletion of talin1 in DCs revealed that talin1 plays a vital role in DC antigen sampling, DC activation and priming of adaptive immunity [80] but not in DC migration and arrival to lymph nodes in vivo [249]. However, genetic deletion of talin1 in skin resident DCs, decreased T cell-mediated DC activation and chemotaxis to lymph nodes upon inflammatory stimuli, hindering skin antimicrobial and immune responses [81]. Although these studies highlight an important function of talin in DCs, the role of DC talin expression in cancer remains elusive. Talin could also affect the innate TME by regulating the function of NK cells and TANs. Talin-depleted NK cells had impaired α_L_β_2_ integrin-mediated adhesion and cytotoxicity [82], whereas talin deficiency in myeloid lineages impaired integrin activation, neutrophil tethering and rolling along the vessel wall and extravasation into inflammatory sites [50,120,156]. Methylation of talin by Ezh2, also decreased adhesion of innate leukocytes, extravasation and organ infiltration in response to inflammatory stimuli [250,251]. Interestingly, a recent study showed that Ezh2 inhibition in mice increased the number of immunosuppressive CD11b^+^Gr1^+^ cells within subcutaneous tumours [252]. In-depth research is yet needed to unravel whether changes in leukocyte differentiation and infiltration within tumours are associated with post-translational modifications in adhesome members.

Additional adhesome components have been implicated to innate immune cell function. FHL2-deficient mice displayed increased DC migration in response to chemokine signalling, hinting at a suppressive role of FHL2 in immune responses [95]. Similarly, conditional depletion of filamin A from myeloid lineages increased neutrophil adhesion and spreading under basal conditions without though affecting immune cell distribution in different organs, adhesion underflow or the generation of reactive oxygen species [101]. However, in an in vivo peritonitis model, filamin A deletion impaired neutrophil chemotaxis, revealing a role of filamin A in inflammatory responses and potentially in cancer [102]. Further studies are required to reveal the function of these proteins in cancer immunity.

Moreover, overexpression of ILK attenuated leukocyte adhesion on ECs [253] and in vitro deletion of ILK in immortalised macrophages decreased their survival [254]. Other proteins of the adhesion network including vinculin and γ-parvin, although they disrupt leukocyte adhesion in vitro [226,243,255], are dispensable for leukocyte function in vivo [86,94]. Interestingly, mutations of the kindlin3 gene have been linked to leukocyte adhesion deficiency syndrome [256], and kindlin3-null mice display impaired integrin activation as well as neutrophil adhesion and arrest on the vascular wall [73,74]. Minimal kindlin3 expression (5% of normal levels) is sufficient for physiological neutrophil function but inadequate for integrin activation and neutrophil responses in injury or inflammation [75]. Collectively, these studies indicate an intricate function of adhesion components and highlight the need for a thorough examination in a cancer setting. Indeed, inducible deletion of kindlin3 from myeloid lineages in a murine melanoma tumour model impaired the ability of nonclassical monocytes to patrol and scavenge tumour cells. Furthermore, it decreased NK cell and DC recruitment and proliferation, thus promoting the development of metastatic lesions in vivo [38].

#### 4.1.3. Actin Regulatory Layer

Several studies have addressed the role of Rho GTPase family in innate cell adhesion and migration. Specifically, the Rho-GTPases members, RhoA, Rac and Cdc42 have been implicated in podosome assembly with diverse effects in migration [69,257,258]. Although ablation of Rac1 or Rac2 from myeloid lineages does not affect macrophage migration [109,110], the Rho/Pyk2/cofilin axis regulates chemokine-induced DC motility [231]. Rac activation by Pyk2 has also been reported to affect NK cell transendothelial migration [69] and the production of reactive oxygen species in neutrophils [259]. Other cytoskeletal regulators, including Wiskott–Aldrich syndrome protein (WASP) and VASP, were found to be required for actin reorganisation during macrophage phagocytosis [260]. Recent data also have interrelated VASP with NK cell-mediated killing [261] and the regulation of polymorphonuclear neutrophil adhesion in response to cytokine stimulation [99].

Collectively, these findings reveal that albeit compensatory mechanisms, actin regulators influence innate immune functions in a cell type-specific manner and could have an impact on cancer. However, which actin-related proteins are essential and how they shape innate immunity during cancer remains to be investigated.

### 4.2. Adaptive Immune Cell Adhesome

Antitumour adaptive immunity relies mainly on the functional diversity of T cell subsets and the crosstalk between T cells, antigen presenting cells (APCs) and other stromal/immune populations [262]. Upon antigen recognition by T cell receptors (TCR) through major histocompatibility complex (MHC) molecules, costimulation and cytokine signals, naïve T cells are conventionally activated and become effector T cells (T_EFF_ cells). Among T_EFF_ cells, CD8^+^ cytotoxic T lymphocytes (CTLs) play a crucial role in tumour surveillance and kill cancer cells through granule exocytosis (granzymes and perforin secretion) and ligand/death receptor-mediated apoptosis [263]. Concomitantly, CD4^+^ T cells shape tumour immune responses at the tumour site and in lymphoid organs either by supporting cancer cell elimination (T helper cells or Th, follicular helper T cells and CD4^+^ CTLs) or by promoting an immunosuppressing TME and tumour growth (mainly CD4^+^ regulatory T cells or Treg cells) [264]. Although the abundance of intratumoural T cells, especially CD8^+^ cells, provides a positive prognostic and predictive value for cancer patients [265], T-cell-mediated tumour clearance is often restricted. Intrinsic (transcriptional, epigenetic, metabolic factors) and extrinsic (interaction with immunosuppressive cells and soluble factors within the TME) parameters renders T cells dysfunctional (anergic, exhausted or senescent), leading to cancer immune evasion [266,267].

Additionally to T cells, B cells are crucial players of cancer adaptive immunity. Both tumour-promoting and antitumorigenic actions have been attributed to B cells [268]. B cells produce tumour reactive antibodies that facilitate tumour killing by NK and CD4^+^ CD8^+^ T cells and phagocytosis by macrophages. On the other hand, B regulating cells and the production of autoantibodies can suppress Th1 and CD8^+^ cytotoxic effects, leading to tumour development. Future studies will define the context-dependent B cell function and explore the use of B cell-based therapies for cancer treatment [269].

#### 4.2.1. Adhesome Signalling

Numerous studies have implicated FAK and Pyk2 in downstream TCR signalling [270], suggesting a central role in T-cell function. Pharmacological inhibition of Pyk2 and RNAi-mediated knockdown in T cells decreased CTL migration [271], cytokine production and CD4^+^ effector responses in vitro [272,273]. Deletion of Pyk2 in vivo did not affect T cell development but impaired T cell activation and generation of short term CD8^+^ T_EFF_ [274]. Moreover, downregulation of Pyk2 and paxillin disrupted microadhesion structures around TCR microclusters at immunological synapses and impaired T-cell activation [275]. Similarly, overexpression of FAK in the Jurkat T cell line increased integrin β_1_ mediated migration, whereas mutation in FAK Y397 phosphorylation site decreased T cell migration [276]. Pharmacological inhibition of FAK in T cells impaired their adhesion, proliferation and activation [64]. In contrast to the in vitro studies, genetic deletion of FAK from CD4^+^ T cells did not affect T cell development, proliferation or activation, possibly due to residual remaining FAK levels [64]. Consistent with this, FAK heterozygous mice also do not develop any T cell defects [63].

Regarding B cell immunity, FAK and Pyk2 phosphorylation affects progenitor B cell adhesion and spreading [277,278,279] and specific deletion of FAK was shown to regulate B cell homing and retention in bone marrow as well as B cell survival and proliferation [65]. Likewise, Pyk2 deficiency impaired B cell migration into the marginal zone of the spleen and humoral responses to thymic antigen responses [70]. However, the involvement of FAK/Pyk2 in tumour adaptive responses is not well-established. An immunomodulatory function of FAK in tumour stroma has been suggested using FAK inhibitors in vivo. Administration of FAK inhibitors or FAK-specific depletion in cancer cells caused tumour regression with increased CD4^+^ and CD8^+^ T cell infiltration and decreased Tregs in vivo. Specifically, CD8^+^ T cell function was correlated with the efficacy of FAK inhibition [30,31]. The inhibitory role of FAK appeared to be cancer-cell specific and was attributed to its kinase activity within the nucleus and the transcriptional regulation of cytokines production [31,280]. Similar changes in intratumoural CTLs were observed in pancreatic tumour-bearing mice treated with FAK inhibitor [22]. This inverse correlation between FAK activity and CD8^+^ infiltration was also reflected in lesions of pancreatic cancer patients [22]. Of note, FAK inhibition combined with chemotherapy or immunotherapy (adoptive T cell transfer or immune checkpoint inhibitors) brought beneficial results in tumour response and survival of treatment-resistant animals [22,30,281]. Although these studies highlight the role of FAK in T cell-mediated cancer immunity and can pave the way for more effective combination therapies against cancer, more comprehensive examination is required to distinguish the importance of FAK function in immune vs. cancer cells.

In addition to FAK and Pyk2, inhibition of paxillin phosphorylation decreased the tumour-targeting function of CD8^+^ T cells, in vitro, and paxillin knockdown reduced the adhesion and spreading of tissue-resident memory T cells (T_RM_) isolated from human lung tumours [282]. T_RM_ cells emerge as a prominent, non-recirculating subset of CD8^+^ T lymphocytes that reside into tissues to rapidly re-encounter with pathogens and cancer [283]. The exact role of T_RM_ cells in antitumoural responses is now beginning to be appreciated [284] and further investigation is needed to understand how adhesome molecules are involved in immunological memory.

Increasing evidence suggests that the GIT/PAK/PIX signalling pathway influences tumour-resident lymphocytes. Mice deficient in αPIX present defective immune responses, with reduced T and B cell proliferation, increased motility and disrupted immune synapse formation and positive selection [71,72]. In a tumour setting, mouse Tregs lacking GIT2 were unable to induce tumour development when compared with wild-type Tregs in an adoptive T cell transfer mouse model [35]. Further in vitro experiments showed that GIT2 knockdown in human Tregs inhibits T_EFF_ proliferation and decreases the surface levels of CD86 costimulatory molecule on B cells. PAK1/2 is also activated in Tregs in a CTLA4/PKCη-dependent manner [35]. Transcriptomics analysis of various CD4^+^ T cell subsets from patient tumours, normal surrounding tissue and peripheral blood revealed that tumour-localising Tregs have unique signatures of upregulated genes including *pak2* [285]. Additionally, mice lacking PAK1 had increased number of spleen T and B cells in contrast to wild type mice developing cancer in the small intestine, distal colon and rectum, indicating that PAK1 might contribute to the tumour eradication by immune cells [34]. Accordingly, PAK1 depletion decreased tumour incidence rates and size in mice developing spontaneous intestinal-colorectal tumorigenesis [34]. It also enhanced CD4^+^ and CD8^+^ T cell recruitment in a mouse model of pancreatic adenocarcinoma, prolonging mouse survival [33].

#### 4.2.2. Adhesion Organisation

Besides adhesome signalling, emerging data demonstrate that adhesion scaffolding proteins have an essential function in adaptive immunity and therefore, could impact cancer development. T cell specific deletion of talin1 displayed defective immunological synapses and transient T cell-APC interactions leading to reduced T cell activation and proliferation [83]. Another in vivo study showed that talin1 is necessary for Treg survival and function [84]. Furthermore, B cell specific ablation of talin1 revealed that, although talin does not affect the maturation of B cells, it is required for humoral responses to T cell dependent antigen and emigration to lymph nodes [85]. The partner of talin, vinculin, also localises to immunological synapses at B cells and is required for B cell stable adhesion and migration [286]. Recent data also implicated filamin A in T cell trafficking to lymph nodes and inflamed skin in vivo [100]. Additionally, T cell specific ILK deletion decreased chemokine-mediated T cell trafficking and survival [90]. A context-dependent role of kindlin3 was revealed through in vivo studies. T cells lacking kindlin3 failed to arrest and transmigrate through blood vessels under normal shear stress conditions. However, an inflamed vasculature, as observed in cancer, provided sufficient integrin ligands to overcome kindlin3 deficiency and rescued the defective extravasation [76]. Taken together, these findings indicate a putative role of integrin-interacting proteins in adaptive antitumour responses.

#### 4.2.3. Actin Regulatory Layer

Genetic ablation studies in mice have revealed that Rho GTPase family members regulate essential T cell functions, including activation, cell division, migration, and adhesion [114,287,288,289]. Mice lacking both Rac1 and Rac2 present defects in T cell development, resulting in decreased CD4^+^ and CD8^+^ thymocytes [111,112]. Other studies also showed that combined Rac1 and Rac2 function regulate T-cell migration and homing to lymph nodes, thymus and spleen [115]. A similar phenotype is observed in the absence of Vav1, Vav2 and Vav3 guanine nucleotide exchange factors of Rac [112]. Likewise, gene targeting of *cdc42* in mice impairs the initial stages of T cell development, proliferation and migration but augments effector and memory T cell differentiation [106,107], indicating a stage-specific role with putative implications in cancer. While other Rho GTPases, including RhoC and RhoG, do not affect T cell function [121,122], specific deletion of RhoA in thymocytes leads to reduced proliferation, survival and differentiation of T cells. Moreover, it abolishes Th1 inflammation responses, resulting in amelioration of autoimmune diseases [118]. These findings imply a putative important, yet unexplored, function in tumour suppressive immunity. Additionally, RhoA affects B cell development but not BCR-dependent proliferation [119]. Similar to RhoA, Cdc42 deficiency and double Rac1/Rac2 deletion have reduced numbers of transitional, marginal zone and follicular B cells and decreased BCR-mediated survival and proliferation [116,289,290]. However, the importance of B cell expression of Rho GTPases in shaping cancer immunity is not explored yet.

Besides Rho-GTPases, evidence exists for other actin regulators, including the ERM family of ezrin-radixin-moesin and the Ena/VASP proteins in T cell function. Specifically, ERM inactivation and removal from the TCR microstructures via the Rac1/Vav signalling axis is important for efficient T cell-APC association [291,292] suggesting that ERM proteins could facilitate immune evasion in cancer. Cytoskeletal remodelling is important for T cell function as is indicated by mutations in WASP that disrupt T cell cytoskeleton in Wiskott–Aldrich syndrome patients [293] and cause defective TCR signalling and activation in Jurkat T cells [294]. Moreover, Ena/VASP combined deficiency in mice is dispensable for T cell development, adhesion and crawling on the vascular endothelium. Nevertheless, recruitment of CD4^+^ activated T cells to inflammatory sites is severely compromised in double Ena/VASP knockout mice due to defective transendothelial migration [98], suggesting a suppressing mechanism of cancer immunity. A more detailed examination is needed to decipher the role of actin regulators in shaping cancer adaptive immunity.

## 5. Conclusions

It is becoming increasingly evident that the tumour stroma is not a bystander but rather a key driver in tumour development. Malignancy is controlled by the cumulative action of the different TME constituents. In the dynamic crosstalk between cancer cells and TME components, cell adhesion to the ECM has a dominant function. Despite the entrenched role of adhesome members in cancer cells, which adhesome proteins are important to regulate TME cellular players, ECs, mural cells, CAFs and immune cells- is poorly understood. While comprehensive approaches dissecting cancer-intrinsic vs. tumour-associated host effects of adhesome components are still lacking, evidence derived from developmental studies could shed light into the functional requirement of adhesome in controlling TME and tumour progression. It is envisaged that the impact of individual adhesome members in cancer could be cell-type-dependent, with both tumour promoting and suppressive effects elicited by the same protein in different TME cellular components and cancer cells. Therefore, gene-targeting approaches are needed to define the requirement of adhesome proteins in specific cell types of TME. Deciphering how cell adhesion organisation and signalling determines tumour stroma architecture and impact on malignancy will set the stage for effective interventions in cancer prognosis and treatment. Specifically, evidence exists about the role of ECM rigidity in defining cancer outcome and studies have revealed that radiation treatment could lead to fibrosis. One avenue for exploration could be to dissect how adhesome members could impact on tumour fibrosis, affect ECM remodelling and define cancer responses to treatment.

## Figures and Tables

**Figure 1 cancers-13-00525-f001:**
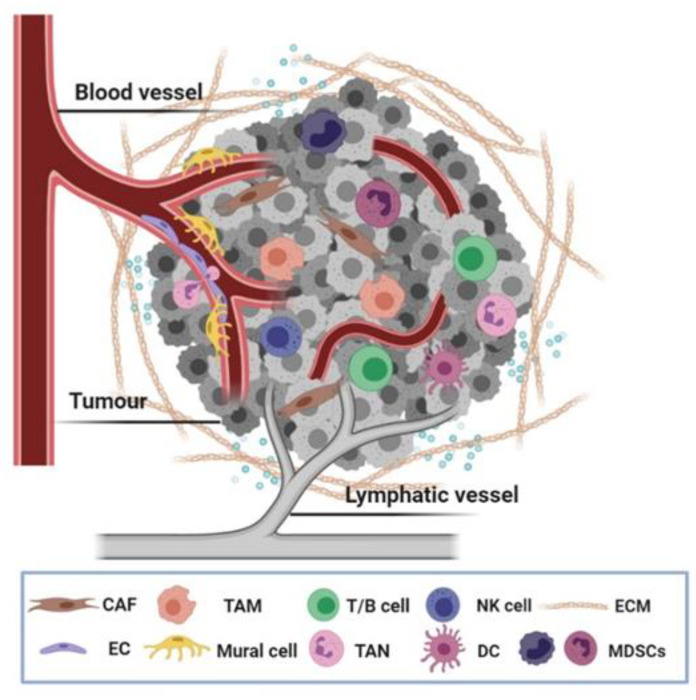
Tumour microenvironment (TME). Besides cancer cells (depicted in grey), distinct cell types and extracellular matrix (ECM) constitute the TME that controls cancer development and progression. Abbreviations: CAF, cancer-associated fibroblast; TAM, tumour-associated macrophages; MDSCs, myeloid-derived suppressor cells; NK cell, natural killer cell; EC, endothelial cell; DC, dendritic cell; TAN, tumour-associated neutrophils; ECM, extracellular matrix.

**Figure 2 cancers-13-00525-f002:**
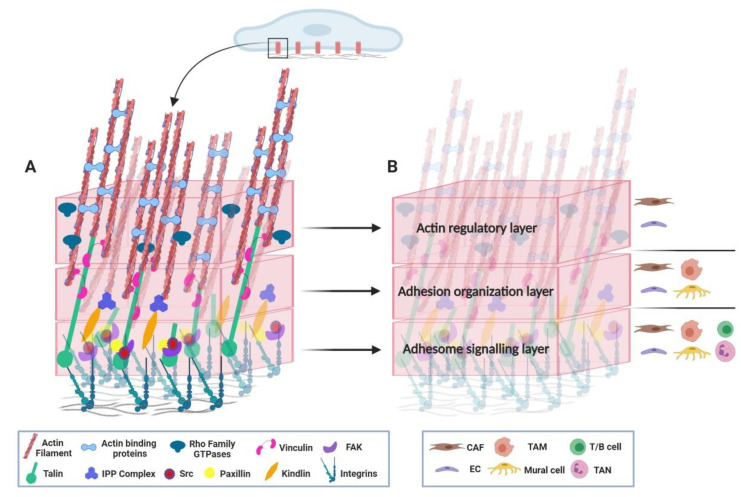
Adhesome in the tumour microenvironment. (**A**) Adhesome molecular architecture. Members of the adhesome, the complex protein network formed at integrin-mediated cell–matrix adhesions (red cylinders at the bottom of the cell), are divided in three levels: the adhesome signalling, the adhesion organisation and the actin regulatory layer (pink boxes). (**B**) Adhesome components and the tumour stroma. The distinct cell types of the TME, with in vivo studied roles of adhesome members are depicted on the right side of each box (layer). Abbreviations: CAF, cancer-associated fibroblast; TAM, tumour-associated macrophage; NK, natural killer; EC, endothelial cell; DC, dendritic cell; TAN, tumour-associated neutrophils; ECM, extracellular matrix.

**Table 1 cancers-13-00525-t001:** Adhesome and cancer: in vivo studies investigating the effect of genetic deletion or pharmacological inhibition of adhesome members in specific tumour stroma cell types.

Molecule	Cell Type	Murine Tumour Model	Intervention	Effect	References
**Adhesome signalling**					
FAK/Pyk2	ECs	Subcutaneous tumours (B16F0 or CMT19T)	*Pdgfb iCreER^T2^; FAK^fl/fl^*	anti-tumourigenic: vessel density (↓), no Pyk2 compensation	[13]
		Orthotopic patient-derived human glioma (DBTRG)	*Tie2 CreER^T2^; FAK^fl/fl^*	anti-tumourigenic: vascular normalisation, no Pyk2 compensation	[14]
		Subcutaneous tumours (B16F0 or CMT19T)	*FAK^het^*	pro-tumourigenic: vessel density (↑)	[15]
		Subcutaneous tumours (B16F10), Metastasis models: I.V or spontaneous metastasis after B16F10 injection	*SCL CreER^T^; FAK^fl/KD^*	metastasis (↓) tumour growth (−)	[16]
		Subcutaneous tumours (B16F0)	*PDGFb iCreER^T^; FAK^fl/fl^; R26FAK^KD/KD^* (FAK kinase-dead mice)	anti-tumourigenic: vessel density (↓), vascular permeability (↓), VE−CAD pY658 levels (↓)	[17]
		Subcutaneous tumours (B16F0)	*PDGFb iCreER^T^; FAK^fl/fl^; R26FAK^DM/DM^* (FAK DM mice: KD with a putatively phosphomimetic Y397E mutation)	anti-tumourigenic: vessel density (↓), vascular permeability (−), VE−CAD pY658 levels (−)	[17]
		Subcutaneous tumours (B16F0 or CMT19T)	*PDGFb iCreER^T^; FAK^fl/fl^; R26FAK^Y397F/Y397F^*	anti-tumourigenic: vessel density (↓)	[18]
		Subcutaneous tumours (B16F0 or CMT19T)	*PDGFb iCreER^T^; FAK^fl/fl^; R26FAK^Y861F/Y8617F^*	no effect: tumour growth (−), vessel density (−)	[18]
	Pericytes	Subcutaneous tumours (B16F0 or LLC)	*PDGFRβ Cre; FAK^fl/fl^*	pro-tumourigenic: vessel density (↑)	[19]
		Subcutaneous tumours (LLC)	*PDGFRβ Cre; FAK^Y397F/Y397F^*	no effect: tumour growth (−), vessel density (−)	[20]
		Subcutaneous tumours (B16F0 or LLC)	*PDGFRβ Cre; FAK^Y861F/Y861F^*	anti-tumourigenic: vessel density (↓)	[20]
	CAFs	Orthotopic xenografts of pancreatic cancer (MPanc-96 or MAD08-608)	Inhibition (FAK/Pyk2 inhibitor: PF-562,271)	anti-tumourigenic: CAFs migration and infiltration (↓)	[21]
		Genetically engineered mice of pancreatic cancer (KPC and KPPC)	Inhibition (FAK inhibitor: VS-4718)	tumour stasis: intratumoural aSMA^+^ and FAP^+^ cells (↓), FAP^+^ cell proliferation (↓)	[22]
		Subcutaneous tumours (HT29)	Inhibition (FAK inhibitor: PF-573,228) or HT29 expressing FAKY^397F^ construct	anti-tumourigenic	[23]
		Syngeneic orthotopic breast (E0771) and pancreatic (TB32048) tumours/MMTV-PyMT mouse model of breast cancer	*FSP Cre^+^; FAK^fl/fl^ mice/MMTV+; FSP Cre+; FAK^fl/fl^*	pro-tumourigenic: CAF−mediated changes in malignant metabolism (↑)	[24]
		Pancreatic cancer cell and fibroblast syngeneic and orthotopic cografting	*FAK^KD^*^(kinase−dead)^ fibroblasts, inhibition (FAK inhibitor: PF562,271)	anti-tumourigenic: tumour growth (−), lung metastasis (↓), fibrosis (↓), M2 macrophage polarisation and migration (↓),	[25]
		MMTV-PyMT mouse model of spontaneous breast cancer	*MMTV-PyMT; Col1a2 CreER; FAK^fl/fl^*	tumour development and growth (−), breast cancer metastasis (↓), exosome amount and functions (↓)	[26]
	TAMs/monocytes	Orthotopic pancreatic tumours (MPanc-96 or MAD08-608)	Inhibition (FAK/PYK2 inhibitor: PF-562,271)	anti-tumourigenic: TAMs (F4/80^+^) migration and infiltration (↓)	[21]
		Genetically engineered mice of pancreatic cancer(KPC and KPPC)	Inhibition (FAK inhibitor: VS-4718)	tumour stasis: intratumoural F4/80^+^ and CD206^+^ macrophages (↓)	[22]
		Syngeneic orthotopic breast tumours (4T1 or MDA-MB-231)	Inhibition (FAK inhibitor: PND-1186)	anti-tumourigenic: metastasis (↓), leukocyte infiltration (↓)	[27]
		Orthotopic breast tumours (4T1)	Inhibition (FAK inhibitors: PF-562271 and PF-573,228)	anti-tumourigenic: intratumoural F4/80^+^ macrophages (↓)	[28]
		MMTV-PyMT mouse model of spontaneous breast cancer	*PyVmT^+/−^;LysM^wt/cre^; FAK^Δmyeloid^*	pro−and anti-tumorigenic: depending on the stage of malignancy, carcinoma stage: NK cells (↓)	[29]
		Subcutaneous tumours (SCC)	Inhibition (FAK inhibitor: BI 853,520)	anti−tumorigenic: PD−L2 surface expression on TAMs (↓)	[30]
	MDSCs/TANs	Genetically engineered mice of pancreatic cancer (KPC and KPPC)	Inhibition (FAK inhibitor: VS-4718)	tumour stasis: recruitment of Gr1+ granulocytes (↓)	[22]
		Subcutaneous tumours (SCC cells)	Inhibition (FAK inhibitor: BI 853,520)	anti-tumorigenic: PD−L2 surface expression on M−MDSCs (↓)	[30]
	CD3^+^/CD4^+^/CD8^+^ T cells	Subcutaneous tumours (SCC cells)	SSC FAK^−/−^ cells, inhibition (FAK inhibitor: VS-4718)	anti-tumourigenic: CD4^+^ and CD8^+^ cell recruitment (↑)	[31]
		Genetically engineered mice of pancreatic cancer (KPC and KPPC)	Inhibition (FAK inhibitor: VS-4718)	tumour stasis: CD4^+^ and CD8^+^ cell recruitment (↑)	[22]
		Subcutaneous tumours (SCC cells)	Inhibition (FAK inhibitor: BI 853,520)	anti-tumorigenic: ICOS surface expression in CD8^+^ cells (↑)	[30]
	Tregs	Genetically engineered mice of pancreatic cancer (KPC and KPPC)	SSC FAK^−/−^ cells, inhibition (FAK inhibitor: VS-4718)	anti-tumourigenic: infiltration of CD4^+^FoxP3^+^CD25^+^ Tregs in the tumour niche (↓)	[31]
		Genetically engineered mice of pancreatic cancer (KPC and KPPC)	Inhibition (FAK inhibitor: VS-4718)	anti-tumorigenic: infiltration of CD4^+^FOXP3^+^ TREGS in the tumour niche (↓)	[22]
		Subcutaneous tumours (SCC)	Inhibition (FAK inhibitor: BI 853,520)	anti-tumorigenic: Tregs (↓)	[30]
Paxillin	ECs	Matrigel plugs implanted subcutaneously (LLC)	siRNA	matrigel plugs: angiogenesis (↑)	[32]
GIT/PIX/PAK	CAFs	Genetically engineered mice of pancreatic cancer (KPC)	*PAK1^−/−^*	anti-tumourigenic: intratumoural aSMA^+^ and desmin^+^ cells (↓)	[33]
	CD3^+^/CD4^+^/CD8^+^ T cells	Genetically engineered mice of pancreatic cancer (KPC)	*PAK1^−/−^*	anti-tumourigenic: CD3^+^, CD4^+^ and CD8^+^ cell recruitment (↑)	[33]
		Genetically engineered mice of intestinal-colorectal cancer APC^Δ14/+^	*PAK1^−/−^*, inhibition (PAK inhibitor PF-3758,309)	anti-tumourigenic: CD3^+^, CD4^+^ and CD8^+^ cell recruitment (↑)	[34]
	Tregs	Subcutaneous tumours (B16F10 or TRAMP-C1)	*GIT2^−/−^*	anti-tumourigenic	[35]
**Adhesion organisation**					
Kindlin2	ECs	Subcutaneous tumours (RM1)	*Kindlin2^+/−^*	anti-tumourigenic: vessel density (↓)	[36]
	CAFs	Subcutaneous coengraftments (SUIT-2 and PSCs)	siRNA	anti-tumourigenic	[37]
Kindlin3	TAMs/monocytes	Intravenous injections of B16F10 cells	*Mx1 Cre^+/−^* (Poly(I:C)); *CX3CR1^gfp/+^; kindlin3^fl/fl^*	pro-tumorigenic: monocyte patrolling (↓), NK cells (↓),	[38]
Talin	ECs	Subcutaneous tumours (B16F0)	*Cdh5 CreER^T2^; talin1^f/L325R^*	anti-tumourigenic: vessel density (↓)	[39]
ILK	ECs	Thyroid (DRO) tumour xenografts	Inhibition (ILK inhibitor: QLT0267)	anti-tumourigenic: vessel density (↓)	[40]
		Orthotopic pancreatic tumour xenografts	Inhibition (ILK inhibitor: QLT0254)	anti-tumourigenic: vessel density (−)	[41]
		Glioblastoma (U87MG) tumour xenografts	Inhibition (ILK inhibitor: QLT0267)	anti-tumourigenic: vessel density (↓)	[42]
		Subcutaneous tumours (PC3)	Inhibition (ILK inhibitor: KP-307-2)	anti-tumourigenic: vessel density (↓)	[43]
**Actin regulatory layer**					
Palladin	CAFs	Orthotopic xenografts of pancreatic cancer (AsPC-1 coimplantation with palladin KD CAFs)	siRNA	anti-tumourigenic	[44]
VASP	CAFs	Subcutaneous tumours (LLC coimplantation with VASP KD MEFs)	siRNA and EVH2 VASP mutant	anti-tumourigenic: intratumoural aSMA^+^ and desmin^+^ cells (↓)	[45]
		Subcutaneous tumours (HT-29 coimplantation with VASP KD HSCs)	siRNA	anti-tumourigenic: intratumoural aSMA^+^ cells (↓)	[46]
Filamin	ECs	Subcutaneous tumours (T241 or B16F0)	*VE-Cad Cre; flnA^fl/fl^*	anti-tumourigenic: vessel density (↓)	[47]
Rac1	ECs	Subcutaneous tumours (B16F0)	*PDGFb iCreER^T^; Rac1^fl/fl^*	no effect: tumour growth (−), vessel density (−)	[48]
		Subcutaneous tumours (Neuro2a cells)	siRNA	anti-tumourigenic: vessel density (↓)	[49]
RhoJ	ECs	Subcutaneous tumours (LLC)	*RhoJ^KO^*	anti-tumourigenic: vessel density (↓)	[50]
		Subcutaneous tumours (B16F0 or LLC)	*Cad5 CreER^T2^; RhoJ^fl/fl^*	anti-tumourigenic: vessel density (↓)	[51]

↓ denotes reduction; ↑ denotes increase; - denotes no difference; Bold, the adhesome layers, as discussed in the text; Orange, adhesome signalling; Blue, Adhesion organization; Green, actin regulatory layer; *italic*, mouse models that have been used in the cited studies.

**Table 2 cancers-13-00525-t002:** Adhesome and physiology: in vivo studies describing the effect of deletion of adhesome members in specific cell types that participate in tumour microenvironment. Abbreviations: HSCs, hematopoietic stem cells.

Molecule	Cell Type	Mouse Model	Effect	References
**Adhesome signalling**				
Paxillin	ECs	siRNA	Retinal angiogenesis (↑), vessel sprouting (↑)	[32]
FAK	ECs	*Tie2 Cre; FAK^fl/fl^*	EC migration (↓), EC proliferation (↓), EC retraction and death (↑), haemorrhage, vessel growth (↓), vessel regression (↑), embryonic lethality (E10.5–E11.5)	[57,58]
		*Tie2 Cre; FAK^flox/KD^*	Vascular permeability (↑), VE-cadherin Y658 phosphorylation (↓), embryonic lethality (E13.5)	[59]
		*End-SCL CreER^T^; FAK^fl/fl^*	No effect-Pyk2 compensation (adult mice)	[60]
	Myeloid lineages → macrophages	*LysM cre; FAK^fl/fl^*	Macrophages: adhesion (↓), migration and invasion (↓), recruitment (↓)	[61]
	Myeloid lineages→ neutrophils	*LysM cre; FAK^fl/fl^*	Adhesion to fibronectin or ICAM-1 (↓), life span (↓), pathogen-killing capability (↓)	[62]
	T cells	*FAK^+/−^*	Normal T cell development	[63]
		*CD4 Cre; FAK^fl/fl^*	Normal T cell development	[64]
	B cells	*CD19 Cre; Fak ^fl/fl^*	Progenitor B and immature B cells (↓), homing to the BM (↓), retention in BM (↓)	[65]
Pyk2	Macrophages	*Pyk2^–/–^*	Macrophages: cell polarisation (↓), cell contractility (↓), size of lamellipodia (↑), migration (↓),	[66]
	Neutrophils	*Pyk2^–/–^*	Neutrophil migration (↓), degranulation (↓)	[67]
	Monocytes	*Pyk2^–/–^*	Number of BM monocyte-lineage cells (↓), Pyk2 promotes the turnover of monocytes at steady state	[68]
	T cells	*Pyk2^–/–^*	Normal T cell development	[66]
		*Pyk2^–/–^*	CD8 T cell activation (↓), LFA-1-dependent CD8 T cell adhesion and migration (↓), CD8^+^ T_EFF_ (↓)	[69]
	B cells	*Pyk2^–/–^*	B cell migration into the marginal zone of spleen (↓), humoral responses to T dependent antigen (↓)	[70]
a-PIX	Immune cells→ T and B cells	*a-PIX^−/−^*	Mature lymphocytes (↓), T cell proliferation (↓), B cell proliferation (↓), B or T motility (↑), disrupted immune synapse	[71]
	T cells	*a-PIX^−/−^*	Thymocytes motility (↑), T cell positive selection (↓)	[72]
**Adhesion organisation**				
Kindlin3	Neutrophils	*kindlin3^−/−^*	Neutrophil: firm adhesion (↓), vascular arrest (↓), recruitment (↓), bleeding, death	[73,74]
	HSCs→ neutrophils	*Mx1 Cre* (Poly(I:C))*; kindlin3^fl/fl^*	Leukocyte adhesion (↓), bleeding, death, 5% protein expression viable	[75]
	T cells	*kindlin3^−/−^, Mx1 Cre* (Poly(I:C)); *kindlin3^fl/fl^, CD4 Cre; kindlin3^fl/fl^*	Thumocyte proliferation (↓), T cell homing to thymus (↓),T cell adhesion at low vascular shear stress (↓)	[76]
Talin1	ECs	*Tie2 Cre; talin1^fl/fl^*	Haemorrhaging, vessel density (↓), small and round ECs, embryonic lethality (E10.5)	[77]
		*Cdh5 CreER^T2^; talin1*	Haemorrhaging of intestine vasculature, death (adult mice)	[78]
		*Cdh5 CreER^T2^; talin1*	Retinal angiogenesis (↓), haemorrhaging, EC proliferation (↓), embryonic lethality	[39]
		*Cdh5 CreER^T2^; talin1^fl/L325R^*	Retinal angiogenesis (↓)	[39]
	Neutrophils, platelets	Rap1 binding deficient talin1 knock-in	Neutrophil adhesion (↓), leukocyte adhesion and extravasation (↓)	[79]
	HSCs → neutrophils	*Mx1 Cre* (Poly(I:C))*; talin1^fl/fl^*	Neutrophil slow rolling and vascular arrest (↓), neutrophil recruitment (↓)	[73]
	Dendritic cells	*CD11c Cre; talin1^fl/fl^*	DC activation (↓), T cell and B cell priming responses (↓)	[80]
		*CD11c Cre; talin1^fl/fl^*	DC migration and activation (↓)	[81]
	NK	*Talin1^−/−^* ES -> differentiate into NK	NK LFA-1-mediated-adhesion (↓), NK cytotoxicity (↓): only retained for selective target cells lacking ICAM-1	[82]
	T cells	*CD4 Cre; talin1^fl/fl^*	T cell activation and proliferation (↓), T cell–APC interactions (transient)	[83]
		*CD4 Cre; talin1^fl/fl^*	T regs (↓), T cell–DCs interactions (↓)	[84]
	B cells	*CD19 Cre; talin1^−/fl^*	Normal B cell development and maturation, humoral responses to T-dependent antigens (↓), B cells homing to lymph nodes and BM (bone marrow) (↓)	[85]
Vinculin	HSCs → neutrophils	*Mx1 Cre* (Poly(I:C))*; Vcl^fl/fl^*	Normal neutrophil recruitment, neutrophil spreading (↓)	[86]
ILK	ECs	*Tie2 Cre; ILK^fl/fl^*	Labyrinthine vascularisation (↓), EC apoptosis (↑), embryonic lethality (E9.5–E10)	[87]
		*PDGFb iCreER^T^; ILK^fl/fl^*	Vessel sprouting (↓), EC proliferation (↓), disruption of the blood–retina barrier	[88]
	vSMCs	*Pdgfrb Cre; ILK ^fl/fl^*	vSMCs: contraction (↑), migration (↓), focal adhesion assembly (↓)	[89]
	T cells	*Lck Cre^+^/ILK^fl/fl^*	T cell chemotaxis (↓), T cell survival (↓)	[90]
Parvin	ECs	*Tie2 Cre; α-pv^fl/fl^*	Haemorrhaging, vessel density (↓), embryonic lethality (E13.5)	[91]
		*Cdh5 CreER^T2^; α-pv^fl/fl^*	Vessel density (↓), vessel sprouting (↓), EC proliferation (↓), vessel regression (↑)	[91]
		*Tie2 Cre; α-pv^fl/fl^; β-pv^−/−^*	Haemorrhaging, vessel branching (↓), vessel diameter (↑), vessel density (↓), embryonic lethality (E10.5–E12.5)	[92]
	vSMCs	*a-pv^−/−^*	vSMCs: contraction (↑), directed migration (↓)	[93]
	Hematopoietic cells	*γ-pv^−/−^*	Normal haematopoiesis	[94]
FHL2	Dendritic cells	*FHL2^−/−^*	DC migration (↑)	[95]
Tensin	ECs	*TNS1^−/−^*	EC migration (↓), EC proliferation (↓)	[96]
**Actin regulatory layer**				
Ena/VASP family	ECs	Ena/VASP triple null	Oedema, haemorrhaging, stress fibre formation (↓), vascular integrity (↓), embryonic lethality (E18.5)	[97]
EVL/VASP	T cells	*EVL^−/−^; VASP^−/−^*	Normal T cell development, activated T cell trafficking (↓), T cell trans endothelial migration (↓)	[98]
VASP	Polymorphonuclear leukocytes or neutrophils (PMNs)	*VASP^−/−^*	Neutrophil/PMNs adhesion (↓)	[99]
Filamin A	T cells	*CD4 Cre; flnA^fl/fl^*	Normal T cell development, Teff flow adhesion (↓), T cell trafficking (↓)	[100]
	Myeloid lineages → neutrophils	*LysM Cre; flnA^fl/fl^*	Neutrophil adhesion (↑), normal adhesion underflow	[101]
	Myeloid lineages → neutrophils	*LysM Cre; flnA^fl/fl^*	Neutrophil recruitment due to inflammatory response (↓)	[102]
Filamin B	ECs	*flnb^−/−^*	Microvascular development (↓)	[103]
Cdc42	ECs	*Tie2 Cre; Cdc42^fl/fl^*	Haemorrhaging, vascular integrity (↓), EC migration (↓), EC proliferation (↓), embryonic lethality (E9–10)	[104,105]
		*Cad5 CreER^T2^; Cdc42^fl/fl^*	Retinal angiogenesis (↓), vessel lumen formation (↓), EC adhesion (↓), embryonic lethality	[104]
	T cells	*Lck Cre; Cdc42^fl/fl^*	T cell development (↓), CD4^+^CD8^+^ double-positive (DP) (↑), CD4^+^ and CD8^+^ single-positive (↓), T cell migration (↓), T cell survival (↓), CD8^+^ Teff (↑), CD8^+^ T memory (↑)	[106]
		*Lck Cre; Cdc42^fl/fl^*	T cell survival (↓), mature CD4^+^ and CD8^+^ T cells in thymus, spleen and lymph nodes (↓), T cell effector/memory (↑)	[107]
Rac1	ECs	*Tie2 Cre; Rac1^fl/fl^*	Vessel density (↓), EC adhesion (↓), EC migration (↓), embryonic lethality (E9.5)	[43]
		*Cad5 CreER^T2^; Rac1^fl/fl^*	Haemorrhaging, vessel density (↓), retinal angiogenesis (↓), embryonic lethality	[108]
	HSCs→ macrophages	*Mx1 Cre* (Poly(I:C)) *;Rac1^fl/fl^*	Macrophages: elongated morphology, normal migration speed and chemotaxis	[109]
		*Mx1 Cre* (Poly(I:C)) *;Rac1^fl/fl^*	Macrophages: trans endothelial migration (↓)	[110]
	HSCs → T cells	*Mx1 Cre* (Poly(I:C)) *;Rac1^fl/fl^*	T cells (↓), B cells (↑), bone marrow lymphopoiesis (↓), CD4^+^CD8^+^ T cells (↓)	[111]
	T cells	*Lck Cre;Rac1^fl/fl^*	Normal T cell development	[111]
		*CD2 Cre;Rac1^fl/fl^*	Normal T cell development	[112]
	B cells	*CD19 Cre;Rac1 ^fl/fl^*	Normal B cell development	[113]
Rac2	Macrophages	*Rac2^−/−^*	Macrophages: normal trans endothelial migration, elongated morphology, adhesion (↓)	[110]
	T cells	*Rac2^−/−^*	Normal T cell development, T lymphocyte migration (↓), chemotaxis (↓), CD4^+^ and CD8^+^ T lymphocytes in spleen, CD8^+^ T lymphocytes in lymph nodes (↑)	[114]
		*Rac2^−/−^*	Normal T cell development	[112]
Rac1 and Rac1	HSCs→ macrophages	*Mx1 Cre* (Poly(I:C))*; Rac1^fl/fl^; Rac2^−/−^*	Trans endothelial migration (↓), normal migration speed and chemotaxis, stellate or elongated morphology	[110]
	HSCs→ T cells	*Mx1 Cre* (Poly(I:C))*; Rac1^fl/fl^; Rac2^−/−^*	Common lymphoid progenitor (CLP, Lin/IL-7Ra) (↓)	[111]
	T cells	*Lck Cre; Rac1^fl/fl^; Rac2^−/−^*	T cell proliferation (↓), T cell survival (↓), adhesion (↓), migration (↓), immature CD4^+^CD8^+^ and mature CD4^+^ in thymus (↓), CD4^+^ and CD8^+^ in spleen (↓)	[111]
		*CD2 Cre; Rac1^fl/fl^; Rac2^−/−^*	T cell development (↓), CD4^+^CD8^+^ double-positive (DP) (↓), CD4^+^CD8^−^ single-positive (4SP) (↓), CD4^−^CD8^+^ single-positive (8SP) (↓)	[112]
		*dLck iCre; Rac1^fl/fl^; Rac2^−/−^*	T cells (↓), T cell chemotaxis (↓), T cell adhesion (↓), homing to secondary lymphoid organs (↓)	[115]
	B cells	*CD19 Cre; Rac1 ^fl/fl^; Rac2^−/−^*	Normal B cell development in bone marrow, B cell development in spleen (↓), B cell proliferation (↓), B cell survival (↓)	[113]
		*CD19 Cre; Rac1 ^fl/fl^; Rac2^−/−^*	B cell development in bone marrow (↓), B cell development in spleen (↓), B cell chemotaxis (↓)	[116]
RhoA	ECs	*Tie2 Cre; RhoA^fl/fl^*	Normal retinal angiogenesis	[117]
		*Cad5 CreER^T2^; RhoA^fl/fl^*	Normal retinal angiogenesis	[117]
	T cells	*Lck Cre; RhoA^fl/fl^*	T cell proliferation (↓), T cell survival n (↓), T cell differentiation n (↓), mature CD4 CD8 T cells in thymus and spleen (↓), trans endothelial migration (↓), Th1 inflammation responses (↓)	[118]
	B cells	*CD19 Cre; RhoA ^fl/fl^*	B cell development (↓)	[119]
	HSCs→ B cells	*Mx1 Cre* (Poly(I:C))*; RhoA^fl/fl^*	B cell development (↓)	[119]
Rhoj	ECs	*Pdgfb iCreER^T2^:Rhoj^fl/fl^*	Delayed retinal angiogenesis, EC motility (↓)	[120]
RhoG, RhoC	B, T cells	*RhoG^−/−^, RhoC^−/−^*	Normal T and B cell development	[121,122]

↓ reduction; ↑ increase; Bold, the adhesome layers, as discussed in the text; Orange, Adhesome signalling; Blue, Adhesion organization; Green, Actin regulatory layer; *italic*, mouse models that have been used in the cited studies.

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
