# Peer review of "The Adhesome Network: Key Components Shaping the Tumour Stroma"

_cancers, 2021, doi:10.3390/cancers13030525_

Round 1

Reviewer 1 Report

Summary

The authors present a highly engaging review where the components of the tumor microenvironment in the context of adhesion is covered. How these cells physically encounter each other and the structural components of the ECM is central to our understanding of all cancers. Special emphasis on metastasis and the mechanisms required through various adhesion interactions are addressed. This review is ambitious and covers a large depth of cellular players with insightful cartoons. Greater depth could be added or small changes to structure of the review could be done to enhance the usefulness to a larger audience. My specific suggestions for improvement are below.

  • Figure 1 is an elegant cartoon, yet lymph and lymphatics are not present, should they be excluded?
  • Table 1 is too long to digest and should be broken into more useful subsets. I suggest at least a tumor -cell adhesions and then maybe immune or possibly even matrix adhesions?
  • Figure 2 is a little confusing and could be broken into more distinct panels? I think the authors are trying to show the different cytoskeleton and mechanical proteins in many distinct cell types. Perhaps just comparing something different like a CAF vs a endothelium would be useful?
  • I think it would be great if the authors could speculate, comment or address how the ‘adhesome’ is affected by different cancer therapies directly or indirectly. I would think the audience would like to know how radiation could cause fibrosis and change the adhesome etc. etc. This may be more than the authors want to discuss but perhaps a little consideration in the conclusion or future perspectives would be good for the novice reader.

Author Response

Point 1: Figure 1 is an elegant cartoon, yet lymph and lymphatics are not present, should they be excluded?

We thank the reviewer for the suggestion. Figure 1 is now redrawn and lymphatics are included.

Point 2: Table 1 is too long to digest and should be broken into more useful subsets. I suggest at least a tumour -cell adhesions and then maybe immune or possibly even matrix adhesions?

We realised that the table 2 was not marked clearly and possible the reviewer thought as table 1 both table 1 and table 2. Changes in the formatting of the tables generated by the conversion of the horizontal to vertical format in the final pdf, i.e., deletion of lines between cell subjects, etc could add to the misunderstanding. We have now marked both table1 and table 2 clearer, adding coloured boxes to distinguish between the different layers of the adhesome. Moreover, we separated the two tables by text that we think will enhance their understanding.

Point 3: Figure 2 is a little confusing and could be broken into more distinct panels? I think the authors are trying to show the different cytoskeleton and mechanical proteins in many distinct cell types. Perhaps just comparing something different like a CAF vs an endothelium would be useful?

We have redesigned the figure 2 to make it more understandable. The first part defines the molecular architecture of the adhesome dividing it in different layers corresponding to the division in the text. The second part depicts the tumour stroma cell types where the role of adhesome components have been investigated in vivo.

Point 4: I think it would be great if the authors could speculate, comment or address how the ‘adhesome’ is affected by different cancer therapies directly or indirectly. I would think the audience would like to know how radiation could cause fibrosis and change the adhesome etc. etc. This may be more than the authors want to discuss but perhaps a little consideration in the conclusion or future perspectives would be good for the novice reader.

Including the response of adhesome members in cancer therapies, is an excellent suggestion but unfortunately, there is no specific data to cite and discuss. Given that integrins are not discussed in this review, how other adhesome components are affected by different cancer therapies in the distinct tumour stroma cell types is currently unknown. Instead, we have cited in the manuscript all relevant information of how adhesion members, affect the response to treatments. This is known only for FAK and immunomodulating treatments that we have discussed extensively lines 516-519 at pg 17 and lines 654-669 at pg 20.

Furthermore, there is no literature to review regarding the link between tumour fibrosis and adhesome members, besides integrins, which are out of the scope of this review. However, we understand the point of the reviewer to discuss more about cancer therapies and specifically, radiation effect on fibrosis and cancer progression. For this, we have added in the concluded paragraph the following statement:

“Specifically, evidence exists about the role of ECM rigidity in defining cancer outcome and studies have revealed that radiation treatment could lead to fibrosis. One avenue for exploration could be to dissect how adhesome members could impact on tumour fibrosis, affect ECM remodelling and define cancer responses to treatment.”

Reviewer 2 Report

In the review, authors summarize the role of selected cell adhesion components in 4 cellular types of TME and focus mainly on cancer-related phenotypes that have been documented by in vivo experiments. As such the review is quite unique and is likely to attract many readers with an interest somewhere at the interface between cell biology, cancer biology and animal physiology.

Major points: Thought integrins play a major role in cell-ECM adhesions, they are highly underrepresented in the review. This is especially evident in the endothelial part of the review. The authors should add at least a paragraph to this section that discusses the role of integrins in the regulation of tumor angiogenesis and the interaction between endothelial cells and tumor cells.

Minor points:

Figure 2: panels with icons of individual cell types along the cross section of cell-ECM adhesions are confusing. This could be misinterpreted as an indication that when a cell type is missing, the corresponding adhesive layer is not present in that cell type.

Lines 158/159: Y397E mutation in FAK is a phosphomimicking mutation. FAK with this mutation, however, cannot be described as constitutively phosphorylated on Y397. More careful wording must be used.

Author Response

Major points: Thought integrins play a major role in cell-ECM adhesions, they are highly underrepresented in the review. This is especially evident in the endothelial part of the review. The authors should add at least a paragraph to this section that discusses the role of integrins in the regulation of tumour angiogenesis and the interaction between endothelial cells and tumour cells.

In this review, we have discussed all the currently available knowledge regarding the role of adhesome components in tumour stroma. To provide a thorough presentation of the known functions of all adhesome components and not selected ones, we decided to exclude integrins because of space constrictions. Integrins have been extensively studied in the tumour stroma and an overview of their role would be an equally sized manuscript. We believe that excellent reviews exist illustrating the role of integrins in cancer progression and we included more citations in the relevant part cited references 52-55: line 85 pg5. Although it could be quite tempting to write about the role of integrins in tumour endothelium, because the corresponding author has been involved in many of these studies, we believe that would create an unbalanced review. It would generate a preference for a particular tumour cell constituent that although it is our favourite cell type studied in our lab, we could not scientifically justify why we include this information.

Minor points:

Figure 2: panels with icons of individual cell types along the cross-section of cell-ECM adhesions are confusing. This could be misinterpreted as an indication that when a cell type is missing, the corresponding adhesive layer is not present in that cell type.

We have redesigned the figure 2 to make it more understandable. The first part defines the molecular architecture of the adhesome dividing it in different adhesome layers corresponding to the division used in the text. The second part depicts the tumour stroma distinct cell types where the role of adhesome components have been investigated in vivo. The related wording of the figure is also changed.

Lines 158/159: Y397E mutation in FAK is a phosphomimicking mutation. FAK with this mutation, however, cannot be described as constitutively phosphorylated on Y397. More careful wording must be used.

We thank the reviewer for this comment. We have now corrected the wording of line 162/163 pg 10: “an endothelial-specific FAK Y397E mutant, that mimics the phosphorylation of FAK Y397”